# Mechanical metamaterials made of freestanding quasi-BCC nanolattices of gold and copper with ultra-high energy absorption capacity

Hongwei Cheng [1,2], Xiaoxia Zhu [1,2], Xiaowei Cheng [3], Pengzhan Cai [3], Jie Liu [1,2], Huijun Yao [1,2], Ling Zhang [3] ✉ & Jinglai Duan [1,2,4] ✉

Nanolattices exhibit attractive mechanical properties such as high strength, high specific strength, and high energy absorption. However, at present, such materials cannot achieve effective fusion of the above properties and scalable production, which hinders their applications in energy conversion and other fields. Herein, we report gold and copper quasi-body centered cubic (quasi-BCC) nanolattices with the diameter of the nanobeams as small as 34 nm. We show that the compressive yield strengths of quasi-BCC nanolattices even exceed those of their bulk counterparts, despite their relative densities below 0.5. Simultaneously, these quasi-BCC nanolattices exhibit ultrahigh energy absorption capacities, i.e., $100 \pm 6$ MJ m$^{-3}$ for gold quasi-BCC nanolattice and $110 \pm 10$ MJ m$^{-3}$ for copper quasi-BCC nanolattice. Finite element simulations and theoretical calculations reveal that the deformation of quasi-BCC nanolattice is dominated by nanobeam bending. And the anomalous energy absorption capacities substantially stem from the synergy of the naturally high mechanical strength and plasticity of metals, the size reduction-induced mechanical enhancement, and the quasi-BCC nanolattice architecture. Since the sample size can be scaled up to macroscale at high efficiency and affordable cost, the quasi-BCC nanolattices with ultrahigh energy absorption capacity reported in this work may find great potentials in heat transfer, electric conduction, catalysis applications.

Metamaterials are rationally structured composites, made of periodically or aperiodically organized building blocks, that manifest extreme properties and exotic functionalities that go beyond constitutive bulk materials[1]. Owing to the unprecedented possibilities enabled by metamaterials, the past decade has witnessed a rapid development of functional diversity including electromagnetic[2], mechanical[3,4], acoustic[5], and others[6,7]. Among these, energy absorption mechanical metamaterials have been the subject of intense interest because they offer exciting opportunities for highly efficient absorption of mechanical energy, which is crucial for several applications[8–10]. In principle, the ideal material for energy absorption should concurrently possess high strength to counteract penetration and superior energy

[1]Institute of Modern Physics, Chinese Academy of Sciences, Lanzhou 730000, China. [2]School of Nuclear Science and Technology, University of Chinese Academy of Sciences, Beijing 100049, China. [3]International Joint Laboratory for Light Alloys (MOE), College of Materials Science and Engineering, Chongqing University, Chongqing 400045, China. [4]Advanced Energy Science and Technology Guangdong Laboratory, Huizhou 516000, China. ✉e-mail: zhangling2014@cqu.edu.cn; j.duan@impcas.ac.cn

**Fig. 1 | Fabrication of gold quasi-BCC nanolattice and morphological characterization. a** Ion irradiation at multiple directions. **b** Chemical etching of ion tracks. **c** Electrochemical deposition of gold. **d** Free-standing quasi-BCC nanolattice. **e** Photograph of a quasi-BCC nanolattice sample. **f** Low-magnification SEM image of gold quasi-BCC nanolattice. **g** SEM image of cross-sectional quasi-BCC nanolattice cut by FIB. **h** High-magnification SEM image of quasi-BCC nanolattice.

absorption capacity to withstand mechanical impact and, favorably, at low weight and/or volume. The energy absorbed by a material is given by the integral of the plateau stress and the failure or densification strain. In most cases, unfortunately, these properties are substantially in contradictory, i.e., high yield or fracture strength is generally gained at the price of low failure strain, and vice versa, which is well exemplified by bulk ceramics. To tackle this problem, metamaterials have been ushered in through ingenious architectural design and material combination, leading to reasonable compromise and providing higher energy absorption capacity[11,12]. For a mechanical metamaterial, its energy absorption capacity is essentially dominated by the material's properties, including size- and microstructure-induced enhancement, and architectural design[9,13]. From the perspective of material's properties, metals possess natural high strength and high ductility and thus are unparallel candidates for the pursuit of high energy absorption capacity. Diameter reduction, especially at the length scale below 200 nm, can bring about further mechanical enhancement, known as "smaller and stronger"[14,15]. The emergence of beam structures, such as nanolattice structures, offers extensive three-dimensional configuration designability for ultra-lightweight materials with ultrahigh stiffness[3,16], large deformability and recoverability[11,17,18], and ultrahigh specific strength[19]. Simultaneously, architectural rationalizations can endow the materials with additional enhancement and stress platform[20], which is hardly available in conventional foam materials, for instance, dealloyed nanoporous metals. Counterintuitively, recent studies have shown that, in comparison with perfect periodical metamaterials, beam offset defects in nanolattice materials have little detriment to their stiffness and strength[21]. Taking these issues into consideration, it is reasonable to speculate that higher energy absorption capacity could be pursued with nanobeams structured metals under properly designed architecture. Up till now, however, such metamaterials have seldom been reported.

Here, we demonstrate nanobeams structured mechanical metamaterials with outstanding strength and energy absorption capacity, afforded by previously unidentified opportunities of three-dimensional (3D) nanowire networks featured with semi-order architecture composed of multi-directional parallelly orientated and randomly positioned solid nanobeams[22]. In other words, we can think of this structure as a full-node-offset BCC nanolattice. For simplicity, we will refer to as quasi-BCC nanolattice. The gold and copper nanobeams in quasi-BCC nanolattices have a dimeter down to 34 nm. Mechanical measurements demonstrate that ultrahigh energy absorption capacity up to $110 \pm 10$ MJ m$^{-3}$ is realized with the copper quasi-BCC nanolattice.

## Results

### Fabrication and characterization of quasi-BCC nanolattices

The gold and copper quasi-BCC nanolattices were prepared by ion track technology[23], as shown in Fig. 1a–d. Firstly, a beam of swift heavy ions irradiates a piece of polycarbonate film at an angle of 45° with respect to the horizontal plane. Further irradiations can be introduced at other directions upon demand, which is considered as an unparallel advantage of ion track technology. The passage of each ion can easily form a damaged straight path, known as ion track (Fig. 1a), and the areal densities of ion tracks are dictated by the irradiation fluence. Then, each track is etched by chemical etching and consequently transformed into a uniform cylindrical channel (Fig. 1b). Gold and copper quasi-BCC nanolattices are electrochemically deposited in the channels (Fig. 1c). Finally, free-standing quasi-BCC nanolattices are obtained by dissolving the template (Fig. 1d). It is seen that the fabrication method enables an independent control over the key parameters and hence a high flexibility for tailoring quasi-BCC nanolattices, i.e., the orientation, density, diameter, number of nanobeam bundles, and the material of interest as well.

We prepared gold quasi-BCC nanolattices with beam diameters of $117 \pm 5$ nm, $86 \pm 4$ nm, $69 \pm 4$ nm, and $34 \pm 2$ nm, and the corresponding relative densities are 0.48, 0.29, 0.20, and 0.49 respectively. A copper quasi-BCC nanolattice with beam diameter of $34 \pm 2$ nm and a relative density of 0.49 was also prepared using the same procedures. The

**Table 1 | Main parameters of the quasi-BCC nanolattices**

| Sample identifier | Area density (cm$^{-2}$) | Angle | Beam diameter (nm) | Relative density | Material |
|---|---|---|---|---|---|
| Au-117 | $7.1 \times 10^8 \times 4$ | 45° | $117 \pm 5$ | 0.48 | gold |
| Au-86 | $7.1 \times 10^8 \times 4$ | 45° | $86 \pm 4$ | 0.29 | gold |
| Au-69 | $7.1 \times 10^8 \times 4$ | 45° | $69 \pm 4$ | 0.20 | gold |
| Au-34 | $7.1 \times 10^9 \times 4$ | 45° | $34 \pm 2$ | 0.49 | gold |
| Cu-34 | $7.1 \times 10^9 \times 4$ | 45° | $34 \pm 2$ | 0.49 | copper |

main parameters of these samples are listed in Table 1. Taking the Au-117 quasi-BBC nanolattice as an example, Fig. 1e–h shows a photograph and the structure of the as-fabricated gold quasi-BCC nanolattice. The critical feature sizes range from a dozen millimeters (whole specimen) to dozens of nanometers (beam diameter), spanning 5 orders of magnitude of length-scale. Figure 1h shows the successful and random connection (full-node-offset) of nanobeams and each nanobeam has a straight and smooth contour.

Figure 2a–d shows the SEM morphology of gold quasi-BCC nanolattices of Au-117, Au-86, Au-69, and Au-34, and Fig. 2e is the SEM image of a copper quasi-BCC nanolattice of Cu-34. Figure 2f is a magnified SEM image of Fig. 2e. For gold and copper quasi-BCC nanolattices with the same area density of nanobeams ($7.1 \times 10^8 \times 4$ cm$^{-2}$) which is determined by the irradiation fluence, as the diameter decreases, the porosity increases and the connectivity between nanobeams becomes weaker (Fig. 2a–c). In other words, good connectivity can be obtained for quasi-BCC nanolattices with a smaller nanobeam diameter when the area density of nanobeams is higher (Fig. 2d–f). At the same area density ($7.1 \times 10^9 \times 4$ cm$^{-2}$) and diameter, the morphologies of gold (Fig. 2d) and copper (Fig. 2e) quasi-BCC nanolattices are almost identical, evidencing the reliability of the employed fabrication method. To examine the microstructure and the purity of our gold and copper quasi-BCC nanolattices, we have performed characterizations of X-ray diffraction (XRD),

high-resolution TEM, backscattered electron SEM, energy disperse x-ray spectra (EDS), and electron energy loss spectroscopy (EELS). Benefiting from the above methods, we have determined that microstructures of our gold and copper quasi-BCC nanolattices are polycrystalline and, within the detection limit of the above techniques, the nanolattices are in high purity and no impurity was detected. Details are illustrated in Supplementary Discussion 1.

### Mechanical properties and mechanisms

The mechanical properties of our quasi-BCC nanolattices were studied by compression tests. Figure 3a shows the SEM snapshots of the deformation behavior of a representative gold quasi-BCC nanolattice sample (Au-69) under coaxial compression, at a prescribed strain rate of about 0.001 s$^{-1}$. It is seen that the quasi-BCC nanolattice undergoes progressive collapse even until 80% of compressive strain (Fig. 3a and the Supplementary Movie 1), which is in stark contrast to the instantaneous collapse observed in "perfect" periodic nanolattices. The stress-strain curves of all samples are displayed in Fig. 3b. Generally, the mechanical response has gone through three stages, i.e., the elastic stage, the plateau stage, and the densification stage. The compressive stiffness $E$ and compressive strength $\sigma$ are plotted versus the relative density $\bar{\rho}$ in Fig. 3c, d. For gold quasi-BCC nanolattices with the same area density ($7.1 \times 10^8 \times 4$ cm$^{-2}$) and different beam diameters, $E$ is proportional to the 2.2 power of the relative density $\bar{\rho}$ (Fig. 3c), it basically conforms to the square relationship predicted by the Ashby formula[1,3,10,24]. Through the above relationship, it is believed that the material-dependent factor can be used to predict the compressive stiffness of quasi-BCC nanolattices with specific relative density and area density. Compressive strength, which scales with the relative density as $\sigma \sim \bar{\rho}^{2.4}$ has been found for different relative densities with the same area density of gold quasi-BCC nanolattices (Fig. 3d), which is quite different from the prediction of the Gibson-Ashby law with $\sigma \sim \bar{\rho}^{1.5}$. The discrepancies between $E$, $\sigma$, and the predictions of Ashby theory should all be greatly related to the offset nodes in the quasi-BCC nanolattice structure. The nodal offset defect results in an increase in

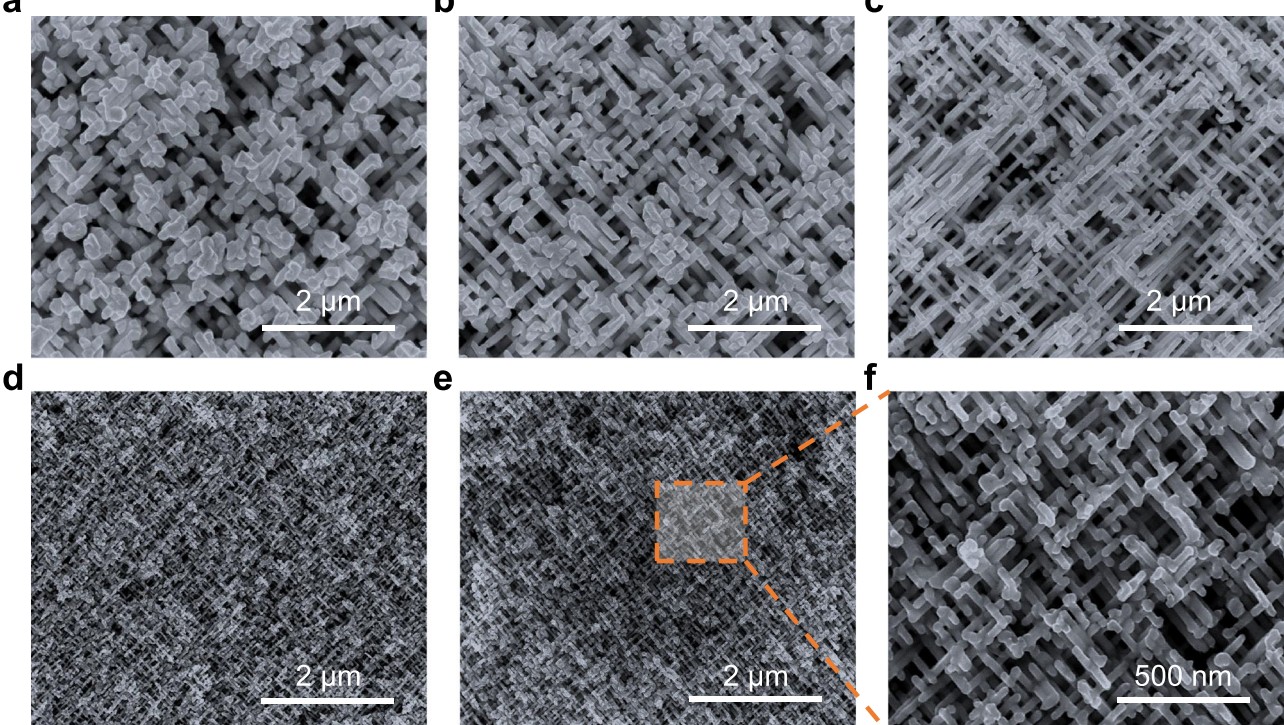

**Fig. 2 | Nanobeam diameter tuning. a–d** SEM images of gold quasi-BCC nanolattices. **a** Au-117 **b** Au-86 **c** Au-69 **d** Au-34, respectively. **e** Copper quasi-BCC nanolattice Cu-34. **f** Magnified SEM image of **e**.

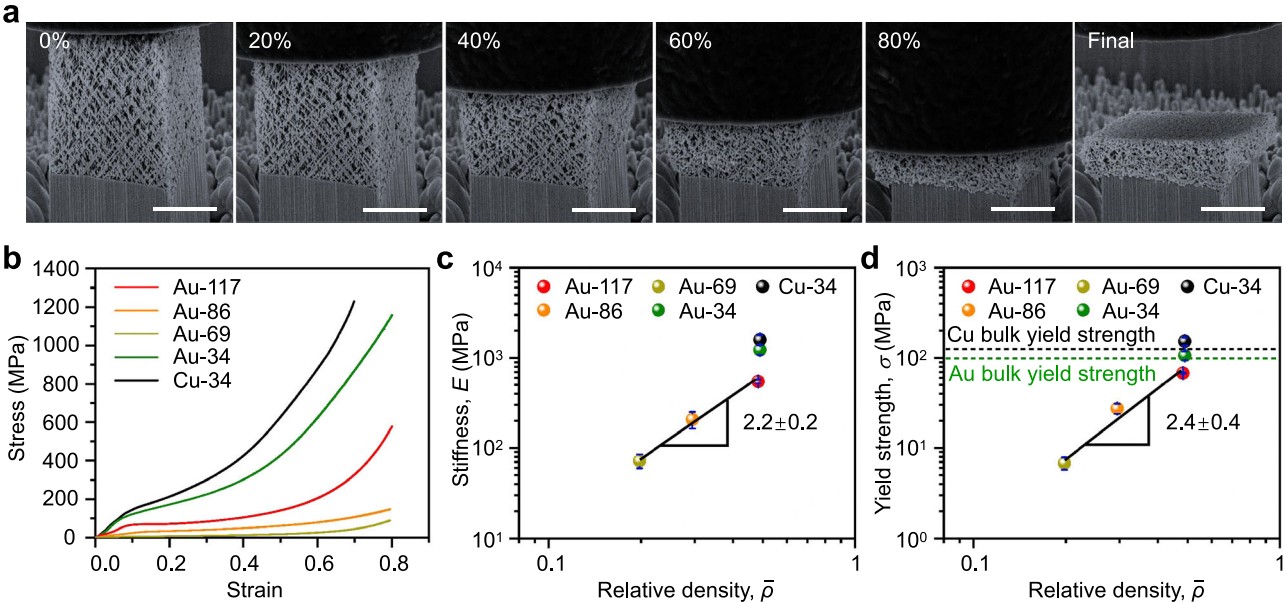

**Fig. 3 | Compression tests on the gold and copper quasi-BCC nanolattices.** **a** SEM snapshots of the deformation process of the sample Au-69. **b** Stress–strain curves of the gold and copper quasi-BCC nanolattices of the samples Au-117, Au-86, Au-69, Au-34, Cu-34, respectively. **c** Compressive stiffness versus relative density and **d** Compressive strength versus relative density of gold and copper quasi-BCC nanolattices. Scale bar 2 μm. The error bars denote the standard deviations of the mean values measured from at least three micropillars. Source data are provided as a Source Data file.

the power exponent, that is, a greater decrease in modulus and strength as the relative density decreases[9,21]. Also noteworthy is that, the strength and the stiffness of the Au-34 quasi-BCC nanolattice can be approximately doubled compared to the Au-117 quasi-BCC nanolattice. The increase in strength can be ascribed to the size effects, which is consistent with the previous studies[9]. However, the increase in stiffness is not from the size effects. Our simulations show that there is no difference in stiffness between two samples (Supplementary Table 3). Comparing with Au-117, the increase in stiffness of Au-34 substantially stems from the difference of surface roughness between two samples. In Supplementary Discussion 2, we illustrate that the surface roughness of the sample Au-117 is larger than that of Au-34. We also show that smaller roughness gives rise to increased stiffness. Therefore, the increase in stiffness of Au-34 should be attributed to the reduction of surface roughness, rather than size effects. To verify the mechanical properties enabled by the quasi-BCC nanolattices, instead of gold, we prepared a copper quasi-BCC nanolattice with the same beam dimeter of 34 nm. Because of naturally higher mechanical strength, the copper quasi-BCC nanolattice shows stronger yield strength than gold, despite the same beam diameter of 34 nm and a relative density of 0.49. Unexpectedly, the yield strengths of both quasi-BCC nanolattices have high values, i.e., 107 ± 11 MPa for gold and 153 ± 15 MPa for copper, and outweigh gold (100 MPa) and copper (130 MPa) bulk counterparts (Supplementary Table 1).

For our 3D gold and copper quasi-BCC nanolattices, nanobeam bending dominates the deformation behavior of this nonrigid topology[5,20,25]. Deformation and failure initiate at the weakest links between the longer nanobeams with large node offset due to the locally concentrated stress applied on these nanobeams[21,26]. As the compression continues, the nanobeams close to the indenter continue to bend and fail, and stress propagates to the constrained bottom region through the quasi-BCC nanolattice structure. The failure of the nanobeam continued to occur in a non-catastrophic bending fashion and the structure continued to densify (Supplementary Movie 1). Thanks to the outstanding plasticity of metallic materials, our quasi-BCC nanolattices can withstand 80% strain without catastrophic collapse. To gain further insights into the quasi-static compression process of the quasi-BCC nanolattices, we performed finite element

simulations (Supplementary Discussion 2) and theoretical analysis (Supplementary Discussion 3). The parameters used in the simulations are listed in Supplementary Table 2. In this work, the influence of offset nodes in the quasi-BCC nanolattice on its mechanical properties was studied through finite element simulation (Supplementary Discussion 2). The results show that the node offset effects have a greater impact on the mechanical properties of our quasi-BCC nanolattices, as comparing with those on the reported octet-truss nanolattices[21]. Moreover, additional simulations have been performed to illustrate the effects of contact state and surface roughness on the stiffness and the strength (Supplementary Discussion 2). We find that the contact state and surface roughness have obvious impacts on the stiffness and a limited influence on the strength. We also show that the compressive strength of a quasi-BCC nanolattice can be well calculated by an equation (Supplementary Discussion 3 and Supplementary Table 4).

The absorbed energy of a material, $U$, is the strain energy, defined as the integral of the corresponding stress-strain response, that is, the area under the stress-strain curve[8]:

$$U = \int_0^{\varepsilon_D} \sigma(\varepsilon)d\varepsilon \qquad (1)$$

where $\varepsilon_D$ is the densification strain. The energy absorption value is a unit volume parameter independent of the nanolattice size. Compared to previous micro/nanolattices, our gold and copper quasi-BCC nanolattices exhibit higher energy absorption capacity (Fig. 4)[13,16–19,27–33]. Although ceramics and carbon nanolattices have high strength and rigidity, they tend to collapse and lose their mechanical properties destructively under a small strain, thus cannot achieve continuous energy absorption, resulting in moderate energy absorption capacity. On the other hand, metallic nanolattices usually have a high non-destructive strain when subjected to force, so that they continuously absorb energy during the deformation process, but the problem is that the current hollow beam metallic nanolattices generally have low strength, which limits their energy absorption capacity. In contrast to hollow beam, our gold and copper quasi-BCC nanolattices consist of nano-sized solid beams and hence exhibit large continuous strains and high compressive strengths when compressed, rendering a high energy

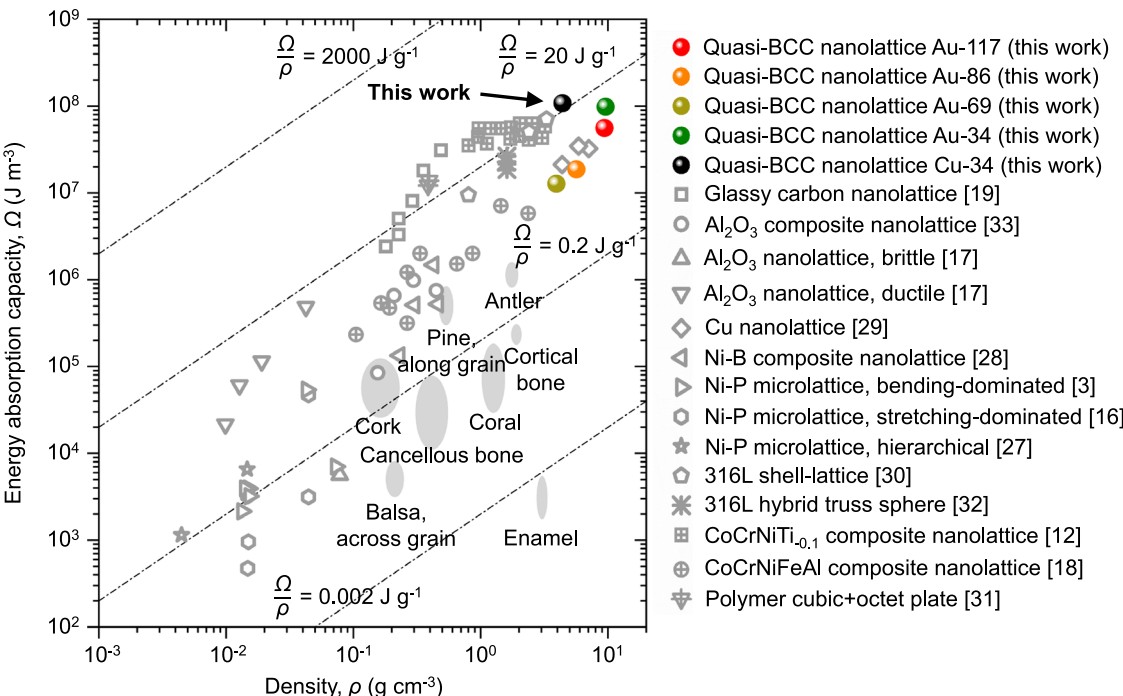

**Fig. 4 | Ashby map of energy absorption per unit volume versus density of gold and copper quasi-BCC nanolattices and previously reported micro/nanolattice metamaterials.** Source data are provided as a Source Data file.

absorption per unit volume (up to $100 \pm 6$ MJ m$^{-3}$ for gold quasi-BCC nanolattice and up to $110 \pm 10$ MJ m$^{-3}$ for copper quasi-BCC nanolattice), surpassing most micro/nanolattices, while being 1–3 orders of magnitude larger than those of natural porous materials with comparable densities. In addition, the copper quasi-BCC nanolattice can achieve an energy absorption per unit mass of 20 J g$^{-1}$, which is commendable for metals with high densities. Such excellent mechanical properties and energy absorption capabilities make gold and copper quasi-BCC nanolattices having great advantages and potential in future multifunctional applications.

### Extendibility of the fabrication method
Macroscopic size is essential for the implementation of metamaterials in future applications. From this perspective, our ion track technology-based fabrication method can generate macroscale flat quasi-BCC nanolattices with high efficiency, compared to previous methods based on 3D printing and self-assembly[34–36]. In contrast to the large lateral size of the quasi-BCC nanolattices, the state-of-art maximum thickness of ion track template is 100 µm which is significantly lower than the centimeter scale thickness fabricated by the self-propagating polymer waveguide technology[37]. In addition, the lower limit of the relative density is estimated through geometric models and finite element simulations in combination with experimental test (Supplementary Discussion 4). With the aid of geometric models and finite element simulations, we show the lower limit of the relative density of our quasi-BCC nanolattices can be as low as 0.01. However, because of the morphological damage induced by the surface tension of dichloromethane during dissolving polycarbonate template, the experimental test shows that the quasi-BCC nanolattice with a relative density of 0.15 starts to lose its structural integrity partially. With the guidance of the geometric model analysis and finite element simulation, it is reasonable to anticipate the quasi-BCC nanolattices with the relative densities lower than 0.2 could be experimentally fabricated by further refining the experimental process.

### Discussion
We provide an in-depth exploration of mechanical gold and copper quasi-BCC nanolattices using experiments, theoretical calculation, and finite element analysis. Our work establishes that gold and copper quasi-BCC nanolattices have excellent compressive strength and energy absorption capacity, which substantially result from the synergy of the naturally high mechanical strength and plasticity of metals, the relevant size reduction-induced mechanical enhancement, and the quasi-BCC nanolattice architecture. We demonstrate that, despite imperfect periodicity, the gold and copper quasi-BCC nanolattices remain slightly lower yet comparable mechanical strength and energy absorption capacity as comparing with perfect nanolattice of similar feature size and relative density. Such a defect-tolerant behavior may promise scalable fabrication methods eligible for future real applications, though structural imperfections are inevitably introduced in the metamaterials. We hope that this work provides some hints for the further design and fabrication of lightweight porous metals with high strength, energy absorption, electrical, and thermal conductivity, and thereby offer promising prospects for realizing high-performance multifunctional applications.

### Methods
#### Preparation
The gold and copper quasi-BCC nanolattices were prepared by electrochemical deposition in the channels of heavy ion track templates. First, the polycarbonate (PC) foils were irradiated by swift heavy ions at the Heavy Ion Research Facility at Lanzhou (HIRFL) with 9.5 MeV per nucleon $^{209}$Bi ions. The thickness of the templates was 30 µm and the fluence of irradiation was $7.1 \times 10^8$ or $7.1 \times 10^9$ cm$^{-2}$ in four directions. Following that, each side of the template was illuminated with UV light for 2 h. The purpose of this step was to make the track etching rate of the template much larger than the bulk etching rate during the etching process, to ensure the uniform channel after etched. Then, the template was placed in 50 °C, 5 M NaOH solution and etched for a certain time to obtain the template with a certain aperture channel. After that,

the etched template was rinsed several times in deionized water immediately and then immersed in deionized water for 5 minutes to remove the remaining etchant from the template to avoid over-etching. A thin layer of gold was sputtered on one template side as an electrochemically deposited cathode and a layer of copper was deposited on the same side to increase the strength of the template using electrolyte consisting of $75\,g\,L^{-1}$ $CuSO_4\cdot5H_2O$ and $30\,g\,L^{-1}$ $H_2SO_4$. The electrolyte used for gold and copper quasi-BCC nanolattices deposition on the other side were $75\,g\,L^{-1}$ $Na_3Au(SO_3)_2$ or $75\,g\,L^{-1}$ $CuSO_4\cdot5H_2O$ and $30\,g\,L^{-1}$ $H_2SO_4$ solution. Last, the PC templates with quasi-BCC nanolattices was placed in dichloromethane ($CH_2Cl_2$) solution to dissolve organic components to obtain gold and copper quasi-BCC nanolattices. All the quasi-BCC nanolattices were stored in ethanol.

The PC template obtained by chemical etching and the electrochemically deposited quasi-BCC nanolattice structure were complementary structures. By weighing out the PC template before and after chemical etching, the relative density of the prepared gold and copper quasi-BCC nanolattice was given by formula (2)[38]:

$$\bar{\rho} = \frac{V - V_e}{V} = \frac{M - M_e}{M} \qquad (2)$$

where $V$ is the overall volume of the template before etching; $V_e$ is the volume of the template after etching; $M$ is the overall mass of the template before etching; $M_e$ is the mass of the template after etching.

### Morphological and chemical characterization
The morphology and crystallinity of gold and copper quasi-BCC nanolattices were analyzed by XRD (Rigaku D/MAX2200pc, Cu K$\alpha$ radiation, $\lambda = 1.54\,Å$), scanning electron microscopy (SEM, FEI Nano-SEM 450, acceleration voltage 15 kV), and transmission electron microscopy (TEM, FEI Tecnai G$^2$ F20, acceleration voltage 200 kV).

### Focused ion beam
The mechanical samples were obtained by cutting centimeter-scale quasi-BCC nanolattices with a focused ion beam system (FIB, FEI, Helios NanoLab 600i). The gold and copper quasi-BCC nanolattices were first pre-cut with a Ga ion beam current of 65 nA at a vacuum of $10^{-6}$ Pa. The micropillars were then finely cut into cubes of approximately $10 \times 10 \times 10\,\mu m^3$ with a Ga ion beam current of 2.5 nA.

### Mechanical characterization
In situ nanocompression testing in SEM (Zeiss Auriga) was performed using a PI 88 (Hysitron) system with a flat indenter (~20 μm diameter). The sample platform was tilted 15° to view live compressed images. The displacement control mode was applied to the micropillar compression with a loading rate of 10 nm s$^{-1}$. The compressive strength $\sigma$ of the structure was defined as the stress at which the structure yields, and the Young's modulus $E$ was determined as the maximum slope of the corresponding stress-strain curve. The densification strains were determined according to Ref. [39], that is, the corresponding strain when the stress rises steeply[39]. At least three micropillars were selected for each sample for repeated compression tests.

### Simulation
Nonlinear simulations were performed for compressive properties of the gold quasi-BCC nanolattices using the explicit solver of the commercial software Abaqus. The Monte Carlo technique was adopted to simulate planar distribution of ion tracks, i.e., the $X$ and $Y$ coordinates of each ion track are generated by a random number generator[40]. In building a simulation model, a tilted beam was placed at the generated $X$ and $Y$ coordinates of each ion track and ultimately formed a 3D quasi-BCC model. Due to limited computational resources, the volumes of models in finite element simulations are $1 \times 1 \times 1\,\mu m^3$ for the area

density of $7.1 \times 10^8 \times 4\,cm^{-2}$ and $0.3 \times 0.3 \times 0.3\,\mu m^3$ for the area density of $7.1 \times 10^9 \times 4\,cm^{-2}$. The relative density difference between the established model and the actual sample was less than 5%. The models were divided into tetrahedral meshes by hyper mesh software, and then imported into Abaqus finite element simulation software for simulation, the ratio of kinetic energy to internal energy must be less than 5% throughout the simulation to ensure quasi-static compression. All nanobeams were rigidly connected, the bottom nodes had zero degrees of freedom in the horizontal and vertical directions, and the top nodes were in contact with the rigid body indenter. The contact attribute of the whole model during compression was set to "all with self", hard contact and separation after contact was allowed. Use the quality scaling feature in the software to ensure the reliability and accuracy of finite element simulations, while increasing simulation speed and saving time.

### Reporting summary
Further information on research design is available in the Nature Portfolio Reporting Summary linked to this article.

## Data availability
The raw data that support the results of this study are available in the source data. The source data used in this study are also available in the Science Data Bank database (https://www.scidb.cn/en/detail?dataSetId=ce57ed50bfa44d1d99adc2cceb404bf9). Source data are provided with this paper.

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

## Acknowledgements

This work was supported by the National Natural Science Foundation of China (Grant No. U1932210, H.C., J.D.) and the Key Research Program of Frontier Sciences, CAS (Grant No. QYZDB-SSW-SLH010, H.C., J.D.). The authors thank the HIRFL staff and Prof. Weiqing Yang and Prof. Youmei Sun for providing ion beams and the assistance with the simulations, Prof. Zhi Qin's group for performing the XRD characterization, and Prof. Yong Peng's group for the assistance with the EELS measurement.

## Author contributions

J.D. conceived the idea. H.C. and J.D. designed the experiments. H.C. have performed various tasks including membrane sample preparation, characterization, performance testing, and finite element simulations. H.C., X.Z., H.Y., J.L., and J.D. irradiated samples. X.Z. was involved in transmission electron microscopy characterization. L.Z., X.C., and P.C. performed in situ mechanical property tests. All authors discussed the results. H.C., X.Z., L.Z., and J.D. contributed to the writing and revision of the manuscript.

## Competing interests

The authors declare no competing interests.
