## [Peer Review File · Nature Communications]

Mechanical metamaterials made of freestanding quasi-BCC nanolattices of gold and copper with ultra-high energy absorption capacityReviewers' Comments:

Reviewer #1:

Remarks to the Author:

This paper demonstrates a new class of meta-materials. The method of ion track etching was used to make nanolattice templates which were then electroplated with Gold or Copper. The resulting lattices comprise solid metal struts in contrast to the more widely published nanolattices based on printed polymer lattices that are coated with a metal and result in hollow struts after polymer removal. Due to the solid metal struts on the nano scale a much higher strength is observed resulting in higher energy absorption.

The work is of significance to this field.

Conclusion and claims are supported well.

I see no flaws in data analysis, interpretation and conclusions.

Reviewer #3:

Remarks to the Author:

Here the authors detail an approach for creating stochastic truss nanolattices via ion tracks from heavy ion irradiation of a polycarbonate film that is subsequently etched to create polymer templates. These voided paths are infilled via electroplating of either gold or copper and the remaining polymer template is etched away to create quasi-BCC nanolattices. The area density of these tracks is controlled via the radiation fluence while the beam diameters is controlled via the etching time allowing lattices of relative densities ranging from approximately 0.2-0.5 to be fabricated. Structures are characterized via SEM and TEM and determined to be predominately single-crystalline. Compression experiments of these focus ion beam milled structures reveal they maintain exceptionally long plateau stresses up to densification and the individual members benefit from size-effect enhanced strength that culminates in the highest energy absorption nanolattices fabricated to date. Overall, the manuscript is well structured, and the methodology is sound and well executed. However, many of the relevant details of the experiments and simulations are absent from the methods and the reported characterization of the constituent material is lacking. The discussion of limitations of this fabrication method and results also requires some additional attention. Therefore, this reviewer cannot recommend this manuscript for publication, but the authors should address the following points to further improve the manuscript.

1. In lines 14 and 15 of the abstract, the authors cite they fabricate nanolattices with full-size as small as 34 nm. This appears to refer to the diameter of the beams. Please rephrase accordingly to improve clarity.
2. In line 25, the authors cite the potential for applications, please be specific.
3. Line 63, quasi-nanolattice may give the wrong impression to some readers that the nanoscale size of the lattice members resides loosely in the range of nanoscale instead of strictly as all members are between 34-117 nm in diameter. It's recommended that the structures be referred to as stochastic truss lattices, quasi-BCC lattices, or some other name to better elucidate that the quasi- prefix to the topology and not the size scale.
4. In the paragraph beginning on line 94, the authors detail control over the relative density via the diameter of nanobeams provided a constant radiation fluence. The authors should elaborate on the limitations of ultralow density structures, e.g. nanobeam connectivity in the limit of small bars. Later in the paper, they detail the creation of nanobeam microstructures via a Monte Carlo technique. Do these Monte Carlo schemes provide an estimate of the lower limit on relative density below which connectivity is too sparse that either the authors anticipate the properties will degrade or they will not

be fabricable? An analogous situation that comes to mind is jamming in packed particles systems at particle volume fractions of $\sim 0.63-0.64$.

5. In conjunction with the above point, the thickness of the irradiated polycarbonate films was 30 microns which is relatively thin compared to other "scalable" truss fabrication techniques such as self-propagating polymer waveguides, albeit with different limitations and advantages. The manuscript does not discuss limitations associated with thickness (e.g. penetration depth of the ion beam which is vital for scalability) nor the potential for anisotropic etching of ion tracks as a result of thicker films via the selectivity of the etchant for the damaged vs undamaged polymer regions.

6. In line 103, the authors cite that the microstructure of the quasi-nanolattices was characterized by TEM, and they determined the purity and microstructure from lattice fringes in two TEM micrographs. Do the authors have additional material that details the microstructure and chemical composition such as back-scatter SEM images, energy dispersive x-ray spectra, x-ray diffraction or electron energy loss spectra? TEM is intrinsically limited to small sample regions and a key claim of this paper is scalability which should be supported by characterization of larger regions. Electroplating can include chemical impurities which can drastically alter the strength of the constituent material even in small concentrations and hence the energy absorptive properties of the structures. Moreover, if the microstructure is predominately single crystalline then the lattice mechanical properties would undoubtedly be influenced by the orientation of the constituent microstructure.

7. In line 155 failure should be corrected to fail.

8. Beginning on line 120, the authors describe the stiffness and strength in the context of foam scaling relations. However, these relations are typically applied to low relative density structures where nodal effects are negligible. It may be the case that offset nodes of the stochastic beam arrangements reduces such effects, but this should be commented on.

9. In conjunction with Point 5 above, the reduction in stiffness is attributed to surface roughness, beam waviness and other factors. Surface roughness is also cited in the discrepancy between simulations and experiments. However, it is difficult to see from the included SEM and TEM micrographs these specific features and their relative degrees. The authors should include characterization of the beams to help the reader better understand the frequency and relative degree of these defects. They also fail to mention whether the simulations include contact, how contact is handled within the FEM framework, and the role of contact in the large deformation of the quasi-BCC nanolattices.

10. In lines 134-138, the authors mention that reducing the size of Au members from 117 to 34 nm doubles both the stiffness and strength. While the size effect associated with the increase in strength is not surprising, the increase in stiffness is and the following sentence that this effect is also observed in single nanowires and nanopillars is unsupported by any citation. The authors should include the relevant citations that demonstrate this supposedly well-established effect as well as discuss it directly in the paper potentially in the context of effects of surface roughness and geometric fidelity if that happens to be relevant to the increase in stiffness with decreasing size. In nanowires of metals with passively oxidized surfaces a change in stiffness can be associated with the change in the relative phase fraction of passive oxide to bulk core. However, this effect should be absent in gold given its nobility.

11. In line 142, the authors mention that the strength of the nanolattices surpasses that of their bulk counterparts, but the methods used to determine the yield strength are absent (e.g. 0.2% yield offset).

12. The authors do not provide the densification strains for any of the tested structures. Please cite these within the table in the supplementary information as this is crucial for transparency and as well as the criteria for which it was chosen, e.g. the global energy absorption efficiency maximum as

determined from the stress-strain curve.

13. In line 161, simulations show compressive stiffness is only slightly lower in the non-periodic structures compared to their periodic counterparts, but no specific value is cited. Please be specific.

14. The authors should comment on the reasoning behind fabricating and testing only one copper structure. Copper possesses a native oxide which would undoubtedly affect the scaling relation with relative density, but as mentioned in the manuscript higher strength and stiffness. The choice of constituent material is critical to scalability and gold would hardly be attune to that.

15. In line 241, the formatting for equations is wrong. Please ensure the final manuscript has the correct formatting.

16. The authors are recommended to include a table of each specimen with a unique identifier directly in the manuscript and refer to each structure via its moniker. At times, it was difficult to recall and follow with structure was being discussed. Moreover, the data points in Figures 3 c and d have error bars, but nowhere in the manuscript does it cite how many of each structural configuration were mechanically tested.

17. As mentioned in several points above, the manuscript fails to elaborate on the methods and assumptions used to determine relevant quantities. For this reason, the authors should revise their methods to be more complete.

Point-by-Point Responses to the Reviewers' Comments

We thankfully acknowledge all the reviewers for their constructive and very helpful comments. We have provided the revised manuscript and supplementary information with the changes highlighted in yellow. Please find below our point-by-point responses. (Note: the reviewers' comments are in black; the replies to the comments are in blue; the revision actions are in red.)

Reviewer #1:

This paper demonstrates a new class of meta-materials. The method of ion track etching was used to make nanolattice templates which were then electroplated with Gold or Copper. The resulting lattices comprise solid metal struts in contrast to the more widely published nanolattices based on printed polymer lattices that are coated with a metal and result in hollow struts after polymer removal. Due to the solid metal struts on the nano scale a much higher strength is observed resulting in higher energy absorption.

The work is of significance to this field.

Conclusion and claims are supported well.

I see no flaws in data analysis, interpretation and conclusions.

Reply: We thank the reviewer for the positive judgement.

Reviewer #3:

Here the authors detail an approach for creating stochastic truss nanolattices via ion tracks from heavy ion irradiation of a polycarbonate film that is subsequently etched to create polymer templates. These voided paths are infilled via electroplating of either gold or copper and the remaining polymer template is etched away to create quasi-BCC nanolattices. The area density of these tracks is controlled via the radiation fluence while the beam diameters is controlled via the etching time allowing lattices of relative densities ranging from approximately 0.2-0.5 to be fabricated. Structures are characterized via SEM and TEM and determined to be predominately single-crystalline. Compression experiments of these focus ion beam milled structures reveal they maintain exceptionally long plateau stresses up to densification and the individual members benefit from size-effect enhanced strength that culminates in the highest energy absorption nanolattices fabricated to date. Overall, the manuscript is well structured, and the methodology is sound and well executed. However, many of the relevant details of

the experiments and simulations are absent from the methods and the reported characterization of the constituent material is lacking. The discussion of limitations of this fabrication method and results also requires some additional attention. Therefore, this reviewer cannot recommend this manuscript for publication, but the authors should address the following points to further improve the manuscript.

Reply: We sincerely thank the reviewer for these constructive and insightful comments which are very important for improving our manuscript. Following the reviewer's suggestions, we have carefully revised the manuscript by adding more details of the experiments and simulations, additional characterizations of the constituent materials, and the discussion of limitations of fabrication methods. After these improvements, we feel that the manuscript has been largely strengthened.

1. In lines 14 and 15 of the abstract, the authors cite they fabricate nanolattices with full-size as small as 34 nm. This appears to refer to the diameter of the beams. Please rephrase accordingly to improve clarity.

Reply: This is a good suggestion and we have replaced “full-size” with “diameter of the beams” in the abstract and the main text.

2. In line 25, the authors cite the potential for applications, please be specific.

Reply: We agree to this suggestion. Examples of the potential applications may refer to heat transfer, electric conduction, catalysis. We have included these applications in the revision. Please see line 25, page 2.

3. Line 63, quasi-nanolattice may give the wrong impression to some readers that the nanoscale size of the lattice members resides loosely in the range of nanoscale instead of strictly as all members are between 34-117 nm in diameter. It's recommended that the structures be referred to as stochastic truss lattices, quasi-BCC lattices, or some other name to better elucidate that the quasi- prefix to the topology and not the size scale.

Reply: Thank you for this wonderful suggestion. We totally agree that the term “quasi-nanolattice” may give rise to wrong impression on our metamaterials. We prefer “quasi-BCC lattices” since it can describe the topology more clearly and properly.

We have accordingly modified the relevant contents in the revised manuscript.

4. In the paragraph beginning on line 94, the authors detail control over the relative density via the diameter of nanobeams provided a constant radiation fluence. The authors should elaborate on the limitations of ultralow density structures, e.g. nanobeam connectivity in the limit of small bars. Later in the paper, they detail the creation of nanobeam microstructures via a Monte Carlo technique. Do these Monte Carlo schemes provide an estimate of the lower limit on relative density below which connectivity is too sparse that either the authors anticipate the properties will degrade or they will not be fabricable? An analogous situation that comes to mind is jamming in packed particles systems at particle volume fractions of ~ 0.63 - 0.64 .

Reply: This comment is constructive, insightful, and inspiring. The relative density is a core parameter. It should be very interesting to know that how the mechanical properties behavior for an ultralow density quasi-BCC lattice. This idea deserves to be systematically investigated as a follow-up work.

For a single monolithic quasi-BCC lattice, the lower density limit is basically determined by the connectivity of beams and, in turn, by the areal density, the diameter, and the length of nanobeams (a longer beam has more possibilities to connect to other beams). To elaborate on the lower limit of the relative density, we have carried out the analysis with the aid of geometric models and finite element simulations (Fig. R1) in combination with experimental test (Fig. R2), using gold as the material. The analysis of geometric models and finite element simulations suggest that, the quasi-BCC lattice of the relative density of only 0.01 nearly remains structural integrity, given that it is in the same dimensions of FIB milled pillars, i.e., $10 \times 10 \times 10 \mu\text{m}^3$, see Fig. R1. However, because of the morphological damage induced by the surface tension of dichloromethane during dissolving polycarbonate template, the experimental test shows that the quasi-BCC lattice with a relative density of 0.15 starts to lose its structural integrity partially (Fig. R2). Details are elucidated below.

The analyses of geometric models and finite element simulations were carried out to estimate the lower limit of relative density from the perspective of structural integrity and performance degradation, respectively. For the analyses, the areal density was fixed to be $7.1 \times 10^8 \times 4 \text{ cm}^{-2}$, corresponding to that of the sample Au-117. At this fixed areal density, the relative densities of 0.15, 0.1, 0.05, and 0.01 were selected to analyze the connectivity of beams by choosing beam diameters of 60 nm, 50 nm, 34 nm, and 15 nm, respectively. From the geometric models of relative density of 0.01 (Fig. R1a-c), we found that the connectivity of beams increases with enlarging model volume of lattice. In the case of model

volume of $1 \times 1 \times 1 \mu\text{m}^3$, 25% beams fail to connect to any other beams, namely, the lattice loses its partial structural integrity (Fig. R1a). The percentage becomes to 8% as the model volume increases to $2 \times 2 \times 2 \mu\text{m}^3$ and further reaches 5% for the model volume of $3 \times 3 \times 3 \mu\text{m}^3$. Although we could not enlarge the model volume further due to limited computational resources, it is reasonable to speculate that, at the volume of $10 \times 10 \times 10 \mu\text{m}^3$ which is the case for our real mechanical tests, nearly 100% beams would connect to other beams. Namely, the lattices are monolithic and have their structural integrity, despite the relative density down to 0.01.

Fig. R1 Geometric models and finite element simulations of gold quasi-BCC lattices with ultralow relative densities. **a** Model volume $1 \times 1 \times 1 \mu\text{m}^3$. **b** Model volume $2 \times 2 \times 2 \mu\text{m}^3$. **c** Model volume $3 \times 3 \times 3 \mu\text{m}^3$. **d** Simulated stress-strain curve. **e** Compressive stiffness versus relative density. **f** Compressive strength versus relative density.

In addition to the analysis based on geometric model, we further simulated the mechanical responses of the quasi-BCC lattices with low relative densities. In simulations, the areal density of beams was fixed to be $7.1 \times 10^8 \times 4 \text{ cm}^{-2}$, corresponding to that of the sample Au-117. The relative density of the gold quasi-BCC lattice decreases by reducing the beam diameter. The relative densities of 0.15, 0.1, 0.05, and 0.01 were tested by choosing beam diameters of 60 nm, 50 nm, 34 nm, and 15 nm, respectively. The lattice thickness was chosen to be 1 μm . In finite element simulations, those

beams unconnected to any other beam were manually removed and did not contribute to mechanical responses. The simulated stress-strain curves show that all the lattices have successful mechanical responses (Fig. R1d). Moreover, it is seen that the stiffness and the strength are highly dependent on the relative density (Fig. R1e,f). In summary, the relative density as low as 0.01 nearly keeps the structural integrity and yields certain mechanical strength.

Fig. R2 SEM images of a gold quasi-BCC lattice with a relative density of 0.15. **a** Low-magnification. **b** Magnified image of **a**.

We have also searched the lower limit of the relative density under our experimental conditions. We found that the gold quasi-BCC lattice with a relative density of 0.15 starts to lose its structural integrity partially, which is attributed to the morphological damage induced by the surface tension of dichloromethane solvent during dissolving polycarbonate template. For this sample, the area density is $7.1 \times 10^8 \times 4 \text{ cm}^{-2}$ (consistent with the sample Au-117), and the beam diameter is $60 \pm 3 \text{ nm}$. The SEM images of the quasi-BCC lattice are shown in Fig. R2. It is seen that, although the sample keeps monolithic (Fig. R2a), the surface morphology is partially damaged at the microscopic scale (Fig. R2b). Guided by the analyses of geometric models and finite element simulations, it is reasonable to anticipate the quasi-BCC lattices with lower relative densities below 0.2 could be experimentally fabricated by further refining the experimental process, for example, reducing or eliminating surface tension of solvents using a freeze-drying method reviewed in Supplementary Ref. 12.

In response to the issue about our Monte Carlo technique, this technique is adopted to simulate planar distribution of ion tracks, i.e., the X and Y coordinates of each ion track are generated by a

random number generator. In building a simulation model, a tilted beam was placed at the generated X and Y coordinates of each ion track and ultimately formed a 3D model. Thus, the adopted Monte Carlo technique itself is not able to provide an estimate of the lower limit on relative density.

Relevant contents have been added to the revised manuscript and the revised Supplementary Information. Please see line 199-208, page 11-12; line 267-270, page 14 and Supplementary Discussion 4.

5. In conjunction with the above point, the thickness of the irradiated polycarbonate films was 30 microns which is relatively thin compared to other “scalable” truss fabrication techniques such as self-propagating polymer waveguides, albeit with different limitations and advantages. The manuscript does not discuss limitations associated with thickness (e.g. penetration depth of the ion beam which is vital for scalability) nor the potential for anisotropic etching of ion tracks as a result of thicker films via the selectivity of the etchant for the damaged vs undamaged polymer regions.

Reply: Thanks for pointing out this important issue. In general, the maximum template thickness mainly relies on penetration depth of energetic ions and etchability and etching selectivity of ion tracks in thicker films. In this paper, we have adopted focused ion beam to mill micropillars for mechanical tests. Because of a practical constraint of milling depth, polycarbonate films with a thickness of 30 μm were selected as the template. In fact, much thicker polycarbonate template could be achievable with ion track technology used in the present work. To confirm this, we have tried polycarbonate of 250 μm in thickness irradiated with 19.5 MeV/u ^{129}Xe ions which have calculated penetration depth of 293 μm in it. Using the etching and electrodeposition parameters employed in this work, we have successfully obtained a polycarbonate template with channel length of about 250 μm (Fig. R3a) and gold nanowires electrodeposited in it (Fig. R3b). Such results provide great potential and confidence for the preparation of thicker quasi-BCC lattice. Despite this, the state-of-art maximum thickness of ion track template is still lower than the centimeter scale thickness fabricated by the self-propagating polymer waveguide technology, although its structural characteristic size is in the order of hundreds of micrometers.

In addition, the maximum thickness of ion track template method raised by the reviewer is very interesting and important, not only for our quasi-BCC lattices, but also for other applications such as ultra-long nanowire growth and filtration. Though a record-thickness of polycarbonate template

fabricated by ion track technology is demonstrated here, we think there has limited space to improve because of penetration depth of the ions with limited energy and potentially low etching selectivity of long ion tracks. Thus, the achievable maximum thickness is still hardly comparable to those obtained with methods such as self-propagating polymer waveguides.

Fig. R3 SEM images of a polycarbonate template of 250 μm in thickness and gold nanowires electrodeposited in it. **a** Cross-sectional SEM image of the template. **b** SEM image of liberated gold nanowires electrodeposited in the template.

Relevant contents have been added to the revised manuscript. Please see line 197-199, page 11.

6. In line 103, the authors cite that the microstructure of the quasi-nanolattices was characterized by TEM, and they determined the purity and microstructure from lattice fringes in two TEM micrographs. Do the authors have additional material that details the microstructure and chemical composition such as back-scatter SEM images, energy disperse x-ray spectra, x-ray diffraction or electron energy loss spectra? TEM is intrinsically limited to small sample regions and a key claim is of this paper is scalability which should be supported by characterization of larger regions. Electroplating can include chemical impurities which can drastically alter the strength of the constituent material even in small concentrations and hence the energy absorptive properties of the structures. Moreover, if the microstructure is predominately single crystalline then the lattice mechanical properties would undoubtedly be influence by the orientation of the constituent microstructure.

Reply: Thank you very much for the constructive comment and valuable suggestions. We agree that

the microstructure and the purity are important factors for the mechanical properties of a solid. To address the issues about the microstructure and the purity of our gold and copper quasi-BCC lattices, we have performed all the characterizations you suggested, i.e., X-ray diffraction (XRD), high-resolution TEM, backscattered electron SEM, energy disperse x-ray spectra (EDS), and electron energy loss spectroscopy (EELS). Benefiting from the above methods, we have determined that microstructures of our gold and copper quasi-BCC lattices are polycrystalline and, within the detection limit of the above techniques, the lattices are in high purity and no impurity was detected. Details are illustrated below.

The microstructures of the gold and the copper lattices were evaluated by XRD and TEM. XRD data of either gold or copper quasi-BCC lattices exhibit typical polycrystalline-like patterns of face-centered cubic (FCC) phases, where the peaks of main crystal planes appear and, in each pattern, the (111) crystal plane shows the strongest diffraction intensity (Fig. R4). We have also verified the polycrystalline microstructure by TEM, taking gold as an example (Fig. R5). It is seen that two beams are composed of three grains, confirming the polycrystalline microstructure.

Fig. R4 XRD data of gold and copper quasi-BCC lattices. **a** Gold. **b** Copper.

Fig. R5 High-resolution TEM images of two interconnected gold beams.

The purity of the gold and the copper were examined by backscattered electron SEM (BSE-SEM), EDS, and EELS techniques. The BSE-SEM images of gold and copper lattices have similar morphologies as those of the secondary electron SEM (SE-SEM) images (Fig. R6). Although the BSE-SEM images have lower signal-to-noise ratio, there is no observable contrast difference, reflecting the lattices have high purity at the microscale. This observation is further supported by the EDS analysis (Fig. R7). The EDS data taken from different regions illustrate that our lattices are only composed of pure gold or copper, respectively. The high purity of gold and copper lattices are further consolidated by EELS spectra (Fig. R8), where no other elements were detected. Based on the above results, it is

safe to conclude that our lattices are in high purity.

Fig. R6 SEM images of gold and copper quasi-BCC lattices. **a** Secondary electron SEM image of a gold quasi-BCC lattice. **b** Backscattered electron SEM image taken from the same region shown in **a**. **c** Secondary electron SEM image of a copper quasi-BCC lattice. **d** Backscattered electron SEM image taken from the same region shown in **c**. The areal density and the beam diameter for the gold quasi-BCC lattice are $3.5 \times 10^8 \times 4 \text{ cm}^{-2}$ and $131 \pm 4 \text{ nm}$, respectively. The areal density and the beam diameter for the copper quasi-BCC lattice are $2.1 \times 10^9 \times 4 \text{ cm}^{-2}$ and $86 \pm 4 \text{ nm}$, respectively.

Fig. R7 SEM images and EDS data of gold and copper quasi-BCC lattices. **a** SEM image and corresponding EDS spectra of a gold quasi-BCC lattice. **b** SEM image and corresponding EDS spectra of a copper quasi-BCC lattice.

Fig. R8 EELS spectra of gold and copper quasi-BCC lattices. **a** Gold quasi-BCC lattice. **b** Copper quasi-BCC lattice.

Relevant contents have been added to the revised manuscript and the revised Supplementary Information. Please see line 101-107, page 6 and Supplementary Discussion 1.

7. In line 155 failure should be corrected to fail.

Reply: Thank you for catching this. We made the correction.

Relevant contents have been modified to the revised manuscript. Please see line 155, page 9.

8. Beginning on line 120, the authors describe the stiffness and strength in the context of foam scaling relations. However, these relations are typically applied to low relative density structures where nodal effects are negligible. It may be the case that offset nodes of the stochastic beam arrangements reduces such effects, but this should be commented on.

Reply: We thank the reviewer for pointing out this. Previous studies (e.g. *Adv. Mater.* **29**, 1701850 (2017)) demonstrate that, for a metamaterial whether in low relative density or in high relative density, the node effects play an important role in the mechanical properties. The node connectivity determines the deformation mechanism of lattice materials under pressure, which is mainly divided into bending-dominated deformation (such as body centered cubic structure) and stretching-dominated deformation (such as octahedral lattice structure). Previous studies show that, resulting from node effects, the mechanical stiffness and strength of the tensile dominated deformation structure are higher than those

of the bending dominated deformation structure, given that they are in the same relative density.

The power exponential relationship was applied to describe the mechanical properties of foam materials as a function of the relative density. Later, this relationship was then introduced to lattice metamaterials, although it has different powers. Moreover, a paper cited in the manuscript (Ref. 21) has studied in detail the influence of node offset defects on the stiffness and strength of octet-truss nanolattices prepared by a two-photon lithography direct laser writing process. Through introducing node offsets in a controlled manner, they found that the lattice showed a minimal change ($\sim 22\%$ decrease) in the yield strength and no observable change in the stiffness. Therefore, they concluded that the node offset has limited detrimental influence on the mechanical properties of octet-truss nanolattices.

Fig. R9 Finite element simulations of the mechanical properties of gold quasi-BCC lattices and periodic BCC lattices under same beam diameter and relative density. **a** Compressive stiffness versus relative density and **b** Compressive strength versus relative density.

In this work, the influence of offset nodes in the quasi-BCC lattice on its mechanical properties was studied through finite element simulation. We have evaluated the stiffness and the strength of quasi-BCC lattices and periodic BCC lattices with the same relative densities and beam diameters (Fig. R9 & Table R1). In our case, the influence of node offset effects on the stiffness depends on the relative density. At the relative density of 0.48, the quasi-BCC lattice has an 8% decrease, i.e., from 9474.2 MPa for the periodic lattice to 8730.5 MPa for the quasi-BCC lattice. At the relative density of 0.20, the quasi-BCC lattice has a 57% decrease, i.e., from 1299.5 MPa for the periodic lattice to 738.5 MPa

for the quasi-BCC lattice. For the strength, at the relative density of 0.48, the quasi-BCC lattice has a 47% decrease, i.e., from 126.1 MPa for the periodic lattice to 66.4 MPa for the quasi-BCC lattice. At the relative density of 0.20, the quasi-BCC lattice has a 53% decrease, i.e., from 26.9 MPa for the periodic lattice to 12.7 MPa for the quasi-BCC lattice. To sum up, the node offset effects have a greater impact on the mechanical properties of our quasi-BCC lattices, as comparing with those on the reported octet-truss lattices.

Table R1 Numerical results of finite element simulation of BCC lattices.

Relative density & Beam diameter	Quasi-/Periodic	Stiffness (MPa)	Yield Strength (MPa)
0.48, 117 nm (Au-117)	Quasi-BCC	8730.5	66.4
	Periodic BCC	9474.2	126.1
0.29, 86 nm (Au-86)	Quasi-BCC	2140.3	30.1
	Periodic BCC	2865.2	54.0
0.20, 69 nm (Au-69)	Quasi-BCC	541.0	12.7
	Periodic BCC	1259.5	26.9
0.49, 34 nm (Au-34)	Quasi-BCC	8783.1	129.5
	Periodic BCC	9996.4	242.2

Relevant contents have been added to the revised manuscript and the revised Supplementary Information. Please see line 158-165, page 9 and Supplementary Discussion 2.

9. In conjunction with Point 5 above, the reduction in stiffness is attributed to surface roughness, beam waviness and other factors. Surface roughness is also cited in the discrepancy between simulations and experiments. However, it is difficult to see from the include SEM and TEM micrographs these specific features and their relative degrees. The authors should include characterization of the beams to help the reader better understand the frequency and relative degree of these defects. They also fail to mention whether the simulations include contact, how contact is handled with in the FEM framework, and the role of contact in the large deformation of the quasi-BCC nanolattices.

Reply: This is an important point. Surface roughness, beam waviness, misalignment of nodes, and others are factors that influence the measured stiffness of a metamaterial. In our quasi-BCC lattices, from the SEM image of a FIB-milled pillar (Fig. R10), it is clearly seen that some neighboring protrusions (beam ends) are in different heights and, as a result, form surface roughness. It would be great to give a relative degree of the surface roughness. Unfortunately, unlike the surface roughness of

a nonporous solid material which can be quantitatively evaluated by techniques such as atomic force microscopy, the relative degree of the surface roughness of porous materials, in particular stochastic truss porous materials like ours, are hardly evaluated quantitatively. For the beam waviness, we determined the waviness from a TEM image (Fig. R11). The results show that the beam waviness of beam 1 is 0.5 nm and the beam waviness of beam 2 is 0.6 nm, respectively, both are below 1 nm and less than 1% (beam diameter $69\pm 2\text{nm}$). Thus, we think beam waviness should play a minor role in influencing the stiffness, as comparing with the surface roughness.

Fig. R10 SEM image of a FIB-milled micropillar.

Fig. R11 TEM image of two interconnected gold beams.

Fig. R12 Schematic diagrams of surface contact states of gold quasi-BCC lattices. **a** Surface contact state 1, the upper end of the nanobeam is fully in contact with the indenter, with a maximum contact area 190249 nm^2 , which is also the contact state used in the finite element simulation involved in this paper. **b** Surface contact state 2, a large part of the upper end of the nanobeam is in contact with the indenter, with a contact area 98358 nm^2 . **c** Surface contact state 3, a small part of the upper end of the

nanobeam is in contact with the indenter, with a contact area 34620 nm^2 . **d** Surface contact state 4, because of surface roughness, not all the beams are in contact with the indenter. For the beams in contact with the indenter, the contact area of each beam equals to that of the contact state 3. **e** Compression stiffness versus contact state. **f** Compression strength versus contact state. In these simulations, the areal density ($7.1 \times 10^8 \times 4 \text{ cm}^{-2}$), the relative density (0.20), and the beam diameter (69 nm) are identical to those of Au-69.

To illustrate the effects of contact state and surface roughness on the stiffness and the strength, we have performed additional finite element simulations (Fig. R12). The areal density and the beam diameter of the simulated quasi-BCC lattices are $7.1 \times 10^8 \times 4 \text{ cm}^{-2}$ and 69 nm, respectively, corresponding to those of the sample Au-69. The model volume is $1 \times 1 \times 1 \text{ }\mu\text{m}^3$. Surface contact state 1 represents the highest contact level (contact area 190249 nm^2), namely, the upper ends of beams are fully in contact with the indenter, which is the case involved in this paper. The state 2 (contact area 98358 nm^2) and the state 3 (contact area 34620 nm^2) represent that the upper ends partly contact with the indenter with reduced contact level. In the state 4, about a half of the number of beams are in contact with the indenter at the contact level prescribed in the state 3, and the rest do not contact. Compared to the state 3, the state 4 has surface roughness. One can see that the compressive stiffness degrades from 541.0 MPa to 332.4 MPa, as the contact changes from the state 1 to the state 3 (Fig. R12e). In comparison, the strength degrades from 12.7 MPa to 12.4 MPa, reflecting that the strength is insensitive to the contact state (Fig. R12f). In short, the surface roughness has an obvious impact on the stiffness and a limited influence on the strength.

Complementary to the above simulations, the experimental stress-strain curve may also give some hints about the surface roughness. We found the stress-strain curves of all the samples will go through two parts at the initial stage during compression. The first part has a smaller slope E_1 and the second part has a larger slope E_2 . For the Au-34 sample, the turning point locates at the strain of 0.02. The E_1 and E_2 are 851.5 MPa and 1651.5 MPa, respectively (Fig. R13a). For the Au-117 sample, the turning point locates at the strain of 0.05. The E_1 and E_2 are 464.2 MPa and 700.3 MPa, respectively (Fig. R13b). We think the smaller slope (E_1) of the first part is very likely due to the surface roughness and the larger slope (E_2) is the stiffness of our quasi-BCC lattices. This explanation may be supported by the findings reported previously (Supplementary Ref. 8 & 9). As such, the turning point of strain reflects, to some extent, the surface roughness.

Fig. R13 Stress-strain curves of the samples Au-34 and Au-117. **a** Au-34. **b** Au-117.

Regarding the issue of the contact in the finite element simulation framework, the contact is set to "all with self", allowing hard contact and separation after contact. During the process of large deformation, the nanobeams inside the structure will contact, but will not penetrate, which is the densification process.

In short, both the contact level and the surface roughness have obvious impacts on the stiffness and limited influence on the strength. The setting of contact in the FEM framework is "all with self".

Relevant contents have been added to the revised manuscript and the revised Supplementary Information. Please see line 165-170, page 9; line 275-279, page 15 and Supplementary Discussion 2.

10. In lines 134-138, the authors mention that reducing the size of Au members from 117 to 34 nm doubles both the stiffness and strength. While the size affect associated with the increase in strength is not surprising, the increase in stiffness is and the following sentence that this effect is also observed in single nanowires and nanopillars is unsupported by any citation. The authors should include the relevant citations that demonstrate this supposedly well-established effect as well as discuss it directly in the paper potentially in the context of effects of surface roughness and geometric fidelity if that happens to be relevant to the increase in stiffness with decreasing size. In nanowires of metals with passively oxidized surfaces a change in stiffness can be associated with the change in the relative phase fraction of passive oxide to bulk core. However, this effect should be absent in gold given its nobility.

Reply: Thank you for pointing out this. We apologize for the confusion induced by our previous

expression on the mechanisms, which result in the strength and the stiffness differences between the samples of Au-117 and Au-34. The increase in strength can be ascribed to the size effects, which is not surprising and commonly accepted. However, the increase in stiffness is NOT from the size effects. Our simulations show that there is no difference in stiffness between two samples (Fig. R9 and Table R1). Comparing with Au-117, the increase in stiffness of Au-34 substantially stems from the difference of surface roughness between two samples. In the response to Point 9, we illustrated that the surface roughness of the sample Au-117 is larger than that of Au-34. We also showed that smaller roughness gives rise to increased stiffness. Therefore, the increase in stiffness of Au-34 should be attributed to the reduction of surface roughness, rather than size effects. We agree that the stiffness increase is not from the relative phase fraction of passive oxide to bulk core because gold is a chemically inert material, as the reviewer pointed out. In addition, these results also suggest that there is some space to improve the stiffness by reducing surface roughness via optimizing fabrication parameters.

Relevant contents have been modified to the revised manuscript. Please see line 129-138, pages 7-8.

11. In line 142, the authors mention that the strength of the nanolattices surpass that of their bulk counterparts, but the methods used to determine the yield strength are absent (e.g. 0.2% yield offset).

Reply: Generally, there are two methods that are adopted to determine the yield strength. 1) For a material with obvious yield, the yield strength is determined to be the stress at the yield point. This method is widely employed to determine the yield strength of lattice materials. 2) For a material without obvious yield phenomenon, the yield strength is determined to be the stress where it deviates from the linear relationship region in the stress-strain curve and reaches the specified value, usually 0.2% of the original gauge length. This method is often used to determine the yield strength of solid materials which have no obvious yield.

In our work, we used the first method to determine the yield strength of our quasi-BCC lattices.

Relevant contents have been added to the revised manuscript. Please see line 261-263, page 14.

12. The authors do not provide the densification strains for any of the tested structures. Please cite these within the table in the supplementary information as this is crucial for transparency and as well as the criteria for which it was chosen, e.g. the global energy absorption efficiency maximum as

determined from the stress-strain curve.

Reply: This suggestion is important. We have added a table to the supplementary information (Table R2), including unique identifiers of the quasi-BCC lattices, compression stiffness, compression strength, densification strain, and energy absorption capacity.

Table R2 Numerical results of experimentally tested quasi-BCC lattices. Mean values and corresponding standard deviations are based on at least three measurements.

Sample	Stiffness (MPa)	Yield Strength (MPa)	Densification strain	Energy Absorption Capacity (MJ m ⁻³)
Au-117	604.3±86.3	68.0±1.2	0.51±0.02	57.3±5.5
Au-86	207.4±43.6	27.3±3.5	0.61±0.01	19.3±2.8
Au-69	72.1±12.7	6.8±1.1	0.65±0.01	13.0±2.3
Au-34	1234.9±70.6	107.0±11.5	0.48±0.03	100.3±6.4
Cu-34	1596.3±194.6	153.3±15.3	0.47±0.03	110.1±9.8

Relevant contents have been added to the revised manuscript and the revised Supplementary Information. Please see line 263-265, page 14 and Supplementary Table 1.

13. In line 161, simulations show compressive stiffness is only slightly lower in the non-periodic structures compared to their periodic counterparts, but no specific value is cited. Please be specific.

Reply: We have added a table (Supplementary Table 3) to the supplementary information, including compressive strength and compressive stiffness of periodic and non-periodic lattices extracted from the simulations.

Relevant contents have been added to the revised manuscript and the revised Supplementary Information. Please see line 161-165, page 9, and Supplementary Discussion 2.

14. The authors should comment on the reasoning behind fabricating and testing only one copper structure. Copper possesses a native oxide which would undoubtedly affect the scaling relation with relative density, but as mentioned in the manuscript higher strength and stiffness. The choice of constituent material is critical to scalability and gold would hardly be attune to that.

Reply: Thank you for your comment. To study the mechanical properties of porous metallic materials, gold is usually the first choice because of its chemical inertness. This may be a reason why there are a lot of papers working on porous gold. This is also the main reason why we chose gold as the model material to study the mechanical properties of quasi-BCC lattice. After a parametric study on gold, we

found that the gold quasi-BCC lattice with the beam diameter of 34 nm and the relative density of 0.49 has the highest stiffness, strength, and energy absorption capacity as well. Having demonstrated the mechanical properties of gold quasi-BCC lattice, we would like to examine the applicability of the fabrication method to other materials, further improve the mechanical performances, and consider the scalability and low cost. Because of these reasons, we tested one copper sample using the optimized parameters. One should mention here that, although short term exposure of copper to air has no obvious oxidation, long term exposure could give rise to oxidation. However, this problem could be solved by surface passivation via, for instance, benzotriazole absorption (*J. Phys. Chem. C* **118**, 8667-8675 (2014)).

Relevant contents have been added to the revised manuscript and the revised Supplementary Information. Please see line 101-107, page 6 and Supplementary Discussion 1.

15. In line 241, the formatting for equations is wrong. Please ensure the final manuscript has the correct formatting.

Reply: Thank you for catching this. The formatting has been corrected.

Relevant contents have been modified to the revised manuscript. Please see line 245, page 13.

16. The authors are recommended to include a table of each specimen with a unique identifier directly in the manuscript and refer to each structure via its moniker. At times, it was difficult to recall and follow with structure was being discussed. Moreover, the data points in Figures 3 c and d have error bars, but nowhere in the manuscript does it cite how many of each structural configuration were mechanically tested.

Reply: Thanks for the suggestions. We have made a unique identifier for each sample (Table R3). The error bars in Figures 3c and d denote the standard deviations of the mean values measured from at least three micropillars.

Table R3 Main parameters of the quasi-BCC lattices.

Sample Identifier	Area Density (/cm ²)	Angle	Diameter (nm)	Relative Density	Material
Au-117	$7.1 \times 10^8 \times 4$	45°	117±5	0.48	gold
Au-86	$7.1 \times 10^8 \times 4$	45°	86±4	0.29	gold
Au-69	$7.1 \times 10^8 \times 4$	45°	69±4	0.20	gold
Au-34	$7.1 \times 10^9 \times 4$	45°	34±2	0.49	gold
Cu-34	$7.1 \times 10^9 \times 4$	45°	34±2	0.49	copper

Relevant contents have been added to the revised manuscript. Please see line 92, page 5; line 149-150, page 9.

17. As mentioned in several points above, the manuscript fails to elaborate on the methods and assumptions used to determine relevant quantities. For this reason, the authors should revise their methods to be more complete.

Reply: We are grateful to the reviewer for these helpful suggestions. In the revision, we have added a number of details to the methods and assumptions.

Reviewers' Comments:

Reviewer #3:

Remarks to the Author:

The authors have excellently addressed all previously mentioned points, having improved both the clarity of the paper and bolstered their claims. Therefore this work is recommended for publication.

Point-by-Point Response to the Reviewers' Comments

Reviewer #3 (Remarks to the Author):

The authors have excellently addressed all previously mentioned points, having improved both the clarity of the paper and bolstered their claims. Therefore this work is recommended for publication.

Reply: We appreciate the reviewer for the recommendation.